# Data Consistency for Weakly Supervised Learning

## Abstract

In many applications, training machine learning models involves using large amounts of human-annotated data. Obtaining precise labels for the data is expensive. Instead, training with weak supervision provides a low-cost alternative. We propose a novel weak supervision algorithm that processes noisy labels, i.e., weak signals, while also considering features of the training data to produce accurate labels for training. Our method searches over classifiers of the data representation to find plausible labelings. We call this paradigm data consistent weak supervision. A key facet of our framework is that we are able to estimate labels for data examples low or no coverage from the weak supervision. In addition, we make no assumptions about the joint distribution of the weak signals and true labels of the data. Instead, we use weak signals and the data features to solve a constrained optimization that enforces data consistency among the labels we generate. Empirical evaluation of our method on different datasets shows that it significantly outperforms state-of-the-art weak supervision methods on both text and image classification tasks.

## 1 Introduction

A common obstacle to adoption of machine learning in new fields is lack of labeled data. Training supervised machine learning models requires significant amounts of labeled data. Collecting labels for large training datasets is expensive. For example, certain natural language processing tasks such as part-of-speech tagging could require domain expertise to annotate each example. Manually annotating each example is time consuming and costly.

Practitioners have turned to crowdsourcing (Howe, 2006) and weak supervision (Ratner et al., 2016; 2019; Arachie & Huang, 2019) as alternative means for collecting labels for training data. Weak supervision trains models by using noisy labels that are cheap and easy to define. The noisy supervision—i.e., weak signals—use rules or patterns in the data to weakly label the dataset. Typically, a user will define multiple weak signals for the data using different rules or heuristics. The weak signals are noisy and thus can conflict on their label estimates. Additionally, they can be correlated and make dependent errors that could mislead a model if the weak signals are naively combined by majority voting or averaging. The key task then in weak supervision training is to intelligently aggregate weak signals to generate quality labels for the training data.

In this paper, we propose a novel weak supervision approach that uses input features of the data to aggregate weak signals and produce quality labels for the training data. Our method works by using the features of unlabeled data and the weak signals to define a constrained objective function. We call this paradigm *data-consistency* since the labels produced by the algorithm are a function of the data input features. We define a *label model* that learns parameters to predict labels for the training data, and these labels must satisfy constraints defined by the weak signals. Both the data features and the weak supervision define the space of plausible labelings, unlike prior works (Ratner et al., 2016; 2019; Arachie & Huang, 2020) that only use the weak signals to learn labels for the training data. Once our algorithm generates data consistent labels, these labels can be used to train any end model as if it were fully supervised.

A major advantage of our approach is that by using features of the data, we are able to estimate labels for examples that have low or no coverage by the weak supervision. In practice, users often define weak signals that do not label some data points. In extreme cases, all the weak signals could abstain on some particular

examples. Many existing weak supervision algorithms do not consider these examples for label estimation. Using the features of the dataset enables us to consider fully abstained examples because our algorithm is able to find similarities between covered examples to estimate labels for the abstain examples. Another advantage of our approach is that we do not assume a family of distributions for the weak signals and the true labels. Assumptions about the labelers or dependence of the weak signals are hard to verify in practice and could cause a method to fail when they do not hold. Our framework can use various representations for input features and offers flexibility on the choice of parametric model for our label model. We can choose a label model family that best fits each task. Lastly, our algorithm can be used for both binary and multiclass classification tasks.

## 1.1 Data Consistent Weak Supervision

The principle behind data consistent weak supervision (DCWS) is that we consider noisy weak supervision labels in conjunction with features of the training data to produce training labels that are consistent with the input data. To do this, we optimize a parametric function constrained by the weak supervision. Formally, let the input data be a set of feature descriptors of examples $X = [x_1, \ldots, x_n]$. These examples have corresponding labels $\boldsymbol{y} = [y_1, \ldots, y_n] \in \{0, 1\}^n$, but they are not available in weakly supervised learning. Instead, we have access to $m$ weak supervision signals $\{\boldsymbol{q}_1, \ldots, \boldsymbol{q}_m\}$, where each weak signal $\boldsymbol{q}_i \in [0, 1]^n$ provides a soft or hard labeling of the data examples. Note that the weak signals can abstain on some examples. In that case, they assign a null value $\emptyset$ to those examples. For multiclass or multi-label classification problems where the labels of the data are a matrix of $K$ classes, each individual weak signal makes one-vs-all predictions for only one class and abstains on other classes. This type of weak supervision is common in practice, since it is somewhat unnatural for human experts to provide rules that label multiple classes on multi-classification tasks.

The core of DCWS is the following optimization:

$$\min_{\theta, \xi \geq 0} \quad \underbrace{\left\| f_\theta(X) - \hat{Y}_r \right\|_2^2}_{\text{regularization}} + \underbrace{C \sum_{i=1}^m \xi_i}_{\text{slack penalty}} \tag{1}$$
$$\text{s.t.} \quad \underbrace{\boldsymbol{A} f_\theta(X) \leq \boldsymbol{b} + \boldsymbol{\xi}}_{\text{data consistent constraints}}$$

We describe the components of this optimization the remainder of this section. We first describe the regularization and then we describe how the data consistent constraints are derived.

**Regularization** The learning objective fits a label model $f_\theta(X)$ to be consistent with weak supervision constraints. We provide a prior labeling $\hat{Y}_r$ that the learning algorithm should default to when given freedom. Choosing a default prior labeling serves as a regularization that can prevent overfitting noisy weak signals or underfitting ambiguous ones. We formulate this regularization as a squared distance penalty

$$\left\| f_\theta(X) - \hat{Y}_r \right\|_2^2. \tag{2}$$

In our experiments, we regularize towards majority vote predictions. For comparison, we also show performance of our method in when it trains with uniform regularization and no regularization.

**Weak Supervision Bounds** Various approaches for weak supervision avoid statistical assumptions on the distribution of the labelers' errors. These approaches instead reason about estimates or bounds of error rates $\boldsymbol{\pi} = [\pi_1, \ldots, \pi_m]$ for the weak signals (Arachie & Huang, 2019; 2021; 2020; Balsubramani & Freund, 2015b;a; Mazzetto et al., 2021b). These values bound the expected error of the weak signals, so they define a space of possible labelings that must satisfy

$$\pi_i \geq \mathbb{E}_{\hat{\boldsymbol{y}} \sim \boldsymbol{q}_i} \left[ \tfrac{1}{n} \sum_{j=1}^n [\hat{y}_j \neq y_j] \right] ,$$

which for one-vs-all weak signals can be equivalently expressed as

$$\pi_i \geq \tfrac{1}{n} \left( \boldsymbol{q}_i^\top (\mathbf{1} - \boldsymbol{y}) + (\mathbf{1} - \boldsymbol{q}_i)^\top \boldsymbol{y} \right) . \tag{3}$$

We extend this bound to cover multi-class classification tasks where the weak signals can choose to abstain on the datasets. Our modified bound is

$$
\begin{aligned}
\pi_i &\geq \frac{1}{n_i} \left( \mathbf{1}_{(\boldsymbol{q}\neq\emptyset)} \boldsymbol{q}_i^\top (\mathbf{1} - \boldsymbol{y}_k) + \mathbf{1}_{(\boldsymbol{q}\neq\emptyset)} (\mathbf{1} - \boldsymbol{q}_i)^\top \boldsymbol{y}_k \right) \\
&\geq \frac{1}{n_i} \left( \mathbf{1}_{(\boldsymbol{q}\neq\emptyset)} (\mathbf{1} - 2\boldsymbol{q}_i)^\top \boldsymbol{y}_k + \boldsymbol{q}_i^\top \mathbf{1}_{(\boldsymbol{q}\neq\emptyset)} \right),
\end{aligned}
\tag{4}
$$

where $\boldsymbol{y}_k$ is the true label for the class $k$ that the weak signal $\boldsymbol{q}_i$ labels, $n_i = \sum \mathbf{1}_{(\boldsymbol{q}_i\neq\emptyset)}$, and $\mathbf{1}_{(\boldsymbol{q}_i\neq\emptyset)}$ is an indicator function that returns 1 on examples the weak signals do not abstain on. Hence, we only calculate the error of the weak signals on the examples they label.

As shown by Arachie & Huang (2020), the expected error rates of the weak signals can be expressed as a system of linear equations $\boldsymbol{Ay} = \boldsymbol{b}$, where

$$
\boldsymbol{A}_i = \mathbf{1}_{(\boldsymbol{q}_i\neq\emptyset)} (\mathbf{1} - 2\boldsymbol{q}_i)
$$

is a linear transformation of a weak signal $\boldsymbol{q}_i$. Each entry in the vector $\boldsymbol{b}$ is the difference between the expected error of the weak signal and the sum of the weak signal, i.e.,

$$
\boldsymbol{b} = n_i \pi_i - \boldsymbol{q}_i^\top \mathbf{1}_{(\boldsymbol{q}\neq\emptyset)}.
$$

We can then rewrite the bound in Equation (4) as

$$
\boldsymbol{Ay} \leq \boldsymbol{b}.
\tag{5}
$$

**Label Model**    The bounds described so far only consider the weak signals. Using the weak supervision $\boldsymbol{q}$ and user provided bounds $\boldsymbol{\pi}$, we have constrained the space of possible labeling for the true labels $\boldsymbol{y}$. However, this constrained space can contain label assignments that are unreasonable given the input data. For example, a feasible labeling could give different classifications to two data examples with identical input. To avoid such inconsistencies, we use a parametric model to enforce an additional constraint that the learned labels are consistent with the input features. We estimate $\boldsymbol{y}$ by defining a parametric model $f_\theta(X)$ that reads the data as input and outputs estimated class probabilities. We refer to this model as the *label model*. Our label model is data-consistent by definition because it relates features of the data to the predicted labels. We combine the label model with the weak supervision by finding parameters $\theta$ whose predictions of the training labels satisfy the weak supervision constraints. By directly substituting the output of the label model for the estimated labels, we obtain a constraint on the parameters:

$$
\boldsymbol{A} f_\theta(X) \leq \boldsymbol{b}.
\tag{6}
$$

This form of the constraint accommodates many forms of the label model $f_\theta(X)$. Depending on the task, $f_\theta(X)$ could be a linear or non-linear model. This gives our method a flexibility that enables practitioners to adapt it for different problem domains. Lastly, for our model, $X$ could be the training data or any input representation of the data. In Section 2, we show experiments with different representations for $X$.

**Slack**    The constraints are provided by the weak supervision $\boldsymbol{q}$ and the error bounds $\boldsymbol{\pi}$. If $\boldsymbol{\pi}$ is too tight, finding solutions to the optimization could be infeasible. In contrast, if $\boldsymbol{\pi}$ is too loose, then the weak supervision will not adequately constrain the objective and the label estimation will not incorporate information from the weak signals. In previous related methods (Arachie & Huang, 2019; Balsubramani & Freund, 2015b;a; Mazzetto et al., 2021b), $\boldsymbol{\pi}$ is calculated on a small set of labeled data, but in practice having access to labeled data may not always be possible. When the bounds cannot be accurately estimated, we choose a tight bound by setting $\boldsymbol{\pi} = 0$, and we use a linear slack penalty to adaptively relax the constraints, yielding the constraint $\boldsymbol{A} f_\theta(X) \leq \boldsymbol{b} + \boldsymbol{\xi}$, where $\boldsymbol{\xi}$ is a vector of nonnegative slack variables, and we add to the objective a slack penalty $C \sum_{i=1}^{m} \xi_i$. In this form, setting $\boldsymbol{\pi} = 0$ becomes equivalent to a weighted majority voting of the weak supervision where the weights are supplied by parameters of the label model.

### 1.2 Optimization

We optimize Equation (1) using Lagrange multipliers. This allows for inexpensive updates to the parameters of the model. At convergence, the Lagrangian function finds a local minimum that adequately satisfies the constraints. The Lagrangian form of the objective is

$$
L(\theta, \xi, \boldsymbol{\gamma}) = \left\| f_\theta(X) - \hat{Y}_r \right\|_2^2 + C \sum_{i=1}^m \xi_i
$$
$$
+ \sum_{i=1}^m \gamma_i \left( \boldsymbol{A}_i f_\theta(X) - b_i - \xi_i \right),
\tag{7}
$$

where $\boldsymbol{\gamma}$ is a Karush-Kuhn-Tucker (KKT) multiplier that penalizes violations on the constraints. During training, we use Adam (Kingma & Ba, 2014) to make gradient updates for the data model $f_\theta(X)$. The Lagrange multiplier $\boldsymbol{\gamma}$ and the slack variable $\boldsymbol{\xi}$ are optimized by gradient ascent and descent respectively. Both the slack variable and the Lagrange multiplier are constrained to be non-negative. This optimization scheme converges when the constraints are satisfied.

## 2 Experiments

Weak supervision algorithms are evaluated on (1) the accuracy of the learned labels on the training data (label accuracy), and (2) the performance of the model on unseen data (test accuracy). Typically, weak supervision algorithms follow a two-stage approach where they first estimate training labels and then use these labels to train an end model that makes predictions on unseen data. DCWS can use a one-stage approach since our label model can itself make predictions on test data, however we use a two-stage approach to measure test accuracy in all our experiments in other to ensure fair comparisons with other methods. To evaluate the effectiveness of DCWS, we design three sets of experiments. First, we train DCWS with the data itself on a synthetic data and measure its label and test accuracy. Secondly, we run DCWS with feature embedding on real data and measure its performance. Lastly, we compare the test accuracy of DCWS to weakly-supervised and semi-supervised methods that train models for label aggregation.

On the first and second set of experiments, we compare the label and test accuracy of DCWS to majority voting, other weak supervision approaches and a crowdsourcing baseline.

**Compared Methods** The state-of-the-art methods we compare to are FlyingSquid (Fu et al., 2020), Snorkel MeTaL (MeTaL) (Ratner et al., 2019), regularized minimax conditional entropy for crowdsourcing (MMCE) (Zhou et al., 2015), constrained label learning (CLL) (Arachie & Huang, 2020), and adversarial label learning (ALL) (Arachie & Huang, 2019). We show test accuracy on supervised learning as reference. It is worth noting that ALL was developed for binary classification tasks and uses weak signals that do not abstain, hence we only run ALL on datasets that satisfy these requirements. We compare to additional baselines used by Awasthi et al. (2020) in our second set of experiments.

**Network Architecture and Hyper-parameters** In all our runs of DCWS, the label model is a two layer neural network with dropout. The hidden layer uses a ReLU activation function and the output layer is sigmoid for binary classification and softmax on multiclass datasets. We set the slack penalty $C = 10$. We run CLL with the same bounds as ours, $\boldsymbol{\pi} = 0$, but we run baseline ALL with the true bounds since the constraints in ALL are very sensitive to incorrect bounds.

To test the generalizability of our algorithm on unseen data, we take the labels produced by DCWS and other methods then train an end model to make predictions on a held-out test set. The model is a simple neural network with two 512-dimensional hidden layers and ReLU activation units. We use the Adam optimizer with the default settings for all experiments.

**Computing Resources and Code** All experiments are conducted using a Tesla T4-16Q vGPU with 16 GB RAM and an AMD EPYC 7282 CPU. Our code is provided in the supplementary materials.

| Method | Label Accuracy |
|---|---|
| DCWS | $0.790 \pm 0.013$ |
| FlyingSquid | 0.622 |
| MeTaL | 0.622 |
| MMCE | 0.625 |
| CLL | 0.625 |
| Majority Vote | 0.625 |
| DCWS+ | $\mathbf{0.821} \pm 0.012$ |

(a) Label Accuracies

| Method | Label Accuracy |
|---|---|
| DCWS | $0.843 \pm 0.009$ |
| FlyingSquid | 0.690 |
| MeTaL | 0.668 |
| MMCE | 0.679 |
| CLL | 0.704 |
| Majority Vote | 0.633 |
| DCWS+ | $\mathbf{0.867} \pm 0.007$ |

(b) Test Accuracies

Table 1: Classification accuracies of the different methods on synthetic data. DCWS trains with datapoints that are covered by the weak supervision, while DCWS+ trains with additional data examples that have no weak supervision coverage. We report the mean and standard deviation over three trials.

| Datasets | DCWS | FlyingSquid | MeTaL | MMCE | CLL | ALL | Majority |
|---|---|---|---|---|---|---|---|
| IMDB | $\mathbf{0.811} \pm 0.004$ | 0.736 | 0.736 | 0.573 | 0.737 | - | 0.701 |
| SST-2 | $\mathbf{0.741} \pm 0.001$ | 0.660 | 0.672 | 0.677 | 0.678 | - | 0.674 |
| YELP-2 | $\mathbf{0.841} \pm 0.003$ | 0.780 | 0.772 | 0.685 | 0.765 | - | 0.775 |
| Fashion Mnist (Binary) | $\mathbf{0.976} \pm 0.005$ | 0.951 | 0.951 | 0.952 | 0.952 | 0.952 | 0.868 |

Table 2: Label accuracies of DCWS compared to other weak supervision methods on different text and image classification datasets. We report the mean and standard deviation over three trials. We do not list the standard deviations if they are less than 0.001.

## 2.1 Synthetic Experiment

The aim of this synthetic experiment is to show that DCWS can perform well when the weak signals are noisy and highly dependent. Additionally, we want to show that with additional datapoints that are not covered by the weak supervision, we are able to get performance gains for our method. We construct a synthetic dataset with 32,000 training examples and 8,000 test examples for a binary classification task where the data has 200 randomly generated binary features. Each feature has between 55% to 65% correlation with the true label of the data. We randomly define 10 weak signals where 9 of the signals are close copies of 1 weak signal, 95% overlap between them. That is, one weak signal is copied noisily 9 times by randomly flipping 5% of the labels. The weak signals have 50% coverage on the data and have error rates between $[0.35, 0.45]$. We run experiments in two settings. In the first setting, DCWS and competing baselines train with only datapoints that are covered by at least one weak signal. In the second setting, DCWS trains with all the datapoints in the training data—including data examples that are not covered by any weak supervision. We refer to this setting as DCWS+. Tables 1a and 1b shows the performance of the different methods in these settings.

As seen in the table, DCWS achieved the highest performance on both label and test accuracies compared to competing baselines. It outperforms the next best performing method by over 16% percentage points on label accuracy and over 13% points on test accuracy. This improvement is significant because FlyingSquid and MeTaL explicitly model the dependency structure between the weak signal and use this information to learn their labels. DCWS does not solve for weak signal dependency but rather uses the features of the training data and the weak signal constraint to defend against possible redundancies. Additionally, we see that DCWS+ obtains better performance than even DCWS, showing that it is able to leverage features from additional examples that are not covered by the weak supervision to better inform the model. Other weak supervision methods typically remove uncovered data examples, but DCWS can synthesize information from these examples to learn more effectively.

## 2.2 Real Data

In this section, we describe the datasets we use in the first experiments and the weak supervision we provide to the learning algorithms. For fair comparison, we only consider training examples that are covered by at

| Datasets | DCWS | FlyingSquid | MeTaL | MMCE | CLL | ALL | Majority | Supervised |
|---|---|---|---|---|---|---|---|---|
| IMDB | **0.779**± 0.002 | 0.745 | 0.685 | 0.555 | 0.685 | - | 0.716 | 0.816 |
| SST-2 | **0.731**± 0.004 | 0.681 | 0.666 | 0.682 | 0.686 | - | 0.672 | 0.783 |
| YELP-2 | **0.842**± 0.004 | 0.816 | 0.780 | 0.68 | 0.826 | - | 0.774 | 0.877 |
| Fashion Mnist (Binary) | **0.970**± 0.010 | 0.940 | 0.940 | 0.938 | 0.941 | 0.942 | 0.892 | 0.992 |

Table 3: Test accuracies of DCWS compared to other weak supervision methods on different text and image classification datasets. We report the mean and standard deviation over three trials. We do not report the standard deviations if they are less than 0.001.

least one weak signal. We use the text datasets and weak signals from Arachie & Huang (2020). For the text classification tasks, we use 300-dimensional GloVe vectors (Pennington et al., 2014) features as representation of the data. The image data uses a pre-trained VGG-19 net (Simonyan & Zisserman, 2014) to extract features of the data. The extracted features are then used to train the models. For more information about the datasets and weak signals, please refer to the appendix and our code.

**IMDB**    The IMDB dataset (Maas et al., 2011) is a sentiment analysis dataset containing movie reviews from different movie genres. The classification task is to distinguish between positive and negative user sentiments. We create the weak supervision signals by considering mentions of specific words in the movie reviews. In total we had 10 weak signals, 5 positive and 5 negative.

**SST-2**    The Stanford Sentiment Treebank (SST-2) is another sentiment analysis dataset (Socher et al., 2013) containing movie reviews. Like the IMDB dataset, the goal is to classify reviews from users as having either positive or negative sentiment. We use similar keyword-based weak supervision but with different keywords leading to 14 total weak signals containing 7 positive sentiments and 7 negative sentiments.

**YELP-2**    We used the Yelp review dataset containing user reviews of businesses from the Yelp Dataset Challenge in 2015. Like the IMDB and SST-2 dataset, the goal is to classify reviews from users as having either positive or negative sentiment. We converted the star ratings in the dataset by considering reviews above 3 stars rating as positive and negative otherwise. We used the same weak supervision generating process as in SST-2.

**Fashion-MNIST**    The Fashion-MNIST dataset (Xiao et al., 2017) represents the task of recognizing articles of clothing. The images are categorized into 10 classes of clothing types. We construct a binary classification task by using two classes from the data. Our task is to classify the tops vs. trousers. We define weak signals for the data by choosing 2 images from each class. The chosen images are used as reference data to generate the weak signals and are excluded from the training data. We calculate pairwise cosine similarity between the embedding of the reference images and the images in the training data and use the rounded scores as the weak supervision labels. Each reference image provides a weak signal hence we have 4 weak supervision signals in total for the dataset.

### 2.3    Results on Real Datasets

Tables 2 and 3 show the performance of our method and various baselines on text and image classification datasets. From Table 2, we see that DCWS consistently outperforms alternative label aggregation approaches on both label and test accuracies. On the IMDB dataset, DCWS outperforms the next best performing method on label accuracy by over 7.5% percentage points and over 6% on the SST and YELP datasets. We see similar results on the test set in Table 3. DCWS's performance is even close to that of supervised learning without using any labeled data. The performance of DCWS in these experiments demonstrates the advantage our method gains by considering features of the data.

The data-free approaches—FlyingSquid, MeTaL, MMCE, and CLL—have accuracy scores that are on par or slightly better than majority voting on some of the datasets. The performance of these methods are greatly affected by the low coverage from the weak supervision. As in the results from the synthetic experiments, DCWS is less affected by the low weak supervision coverage. It is able to synthesize information from the features of the data, which enables it to learn better labels for its model. On all experiments, DCWS

| Methods | Datasets | | | | |
|---|---|---|---|---|---|
| | Question (Accuracy) | MIT-R (F1) | YouTube (Accuracy) | SMS (F1) | Census (Accuracy) |
| Only-L | 72.9± 0.6 | 73.5± 0.3 | 90.9± 1.8 | 89.0± 1.6 | 79.4± 0.5 |
| L+Umaj | 71.5± 1.5 | 73.5± 0.3 | 91.7± 1.9 | 92.5± 1.2 | 80.3± 0.1 |
| Noise-tolerant (Zhang & Sabuncu, 2018) | 72.4± 1.1 | 73.5± 0.2 | 92.6± 1.1 | 91.9± 1.2 | 80.4± 0.2 |
| L2R (Ren et al., 2018) | 73.2± 2.1 | 58.1± 1.0 | 93.4± 0.5 | 91.3± 0.8 | 82.3± 0.3 |
| L+Usnorkel (Ratner et al., 2016) | 72.2± 3.0 | 73.5± 0.2 | 93.6± 0.7 | 92.5± 1.5 | 80.4± 0.4 |
| Snorkel-Noise-tolerant | 71.5± 1.6 | 73.5± 0.3 | 92.9± 0.7 | 91.7± 1.5 | 79.6± 0.5 |
| Posterior Reg. (Hu et al., 2016) | 72.1± 1.0 | 73.4± 0.4 | 88.0± 1.9 | 90.8± 1.5 | 78.6± 0.5 |
| ImplyLoss (Awasthi et al., 2020) | **84.6**± 1.5 | 74.3± 0.3 | 94.1± 1.1 | 93.2± 1.0 | 81.1± 0.2 |
| DCWS | 78.7± 0.7 | **75.2**± 0.6 | **94.5**± 0.5 | **95.0**± 0.0 | **82.4**± 0.2 |

Table 4: Comparison of DCWS with the baselines from (Awasthi et al., 2020) on five different datasets. We report standard deviation of DCWS after over trials. The methods with the best accuracy and F1 score on the test data are bold.

outperforms FlyingSquid, which is considered the current state-of-the-art for latent variable weak supervision models. On the Fashion MNIST dataset, ALL is trained with the true bounds of the weak supervision and as such provides an unfair comparison to competing methods, yet DCWS is still able to outperform ALL.

## 2.4 Comparison to Model Training Methods

In the previous set of experiments, we showed comparison of our method to weak supervision approaches that do not consider the data for label aggregation (with the only exception being ALL). In this section, we will provide experiments that compare DCWS to other data-dependent weak supervision and semi-supervised approaches.

**Baselines** The methods we compare against are from Awasthi et al. (2020). We provide additional details in the appendix explaining each baseline.

**Datasets & Weak Supervision** We used the same datasets and weak signals as Awasthi et al. (2020). Additional details are provided in the appendix. We use the authors' own code and data.[1] The datasets are text classification tasks: three that are binary classification datasets and two that are multi-class datasets. The binary class datasets are SMS Spam Classification (Almeida et al., 2011), Youtube Spam Classification (Alberto et al., 2015), and Census Income (Dua & Graff, 2019). The multi-class datasets are Question Classification (Li & Roth, 2002) and MIT-R (Liu et al., 2013). We used the same models from our previous experiments and regularize towards majority vote predictions. As in our previous experiments, we run DCWS with $\pi = 0$ on binary datasets. However, on the multiclass datasets, we calculate $\pi$ on validation data. Awasthi et al. (2020) used the validation data to tune hyper-parameters for their method and the methods they compare against. Table 4 lists the evaluation of DCWS along with the baseline metrics computed by (Awasthi et al., 2020) (Table 2).

**Results** From Table 4, DCWS outperforms competing baselines on all datasets except for Question Classification. We achieve the second best result on this dataset. On the binary datasets, we are able to outperform competing methods without using the validation data for parameter tuning. Additionally, we did not tune our model for all experiments, we used the setting from previous experiments for consistency. The results of the methods on the datasets are close to that of supervised learning, hence the improvement offered by DCWS is only marginal compared to the next best performing method. The results from these experiments show that DCWS performs as well as semi-supervised methods and weak supervision methods that train models to learn labels.

---

[1] https://github.com/awasthiabhijeet/Learning-From-Rules

### 2.5   Ablation Study

Our data consistency weak supervision approach has different components that enable it to achieve higher quality results in our experiments. In this section, we test the different components to by removing one component at a time and keeping other parts constant to measure relative importance. The various tests we run are training DCWS (i) without slack, (ii) with uniform regularization, (iii) without regularization, (iv) without constraints, and (v) without data consistency. Also, we train DCWS by varying the number of cluster labels for the data representation and also varying the slack parameter and model type. Table 5 lists the results from our ablation study.

**Slack and Regularization**    The results indicate that running DCWS without regularization or slack causes a slight drop in performance of the method. This is consistent with our intuition about the roles of the both components. Slack helps us to adaptively loosen the bounds, while regularization makes our method more stable. Interestingly, using uniform regularization performs as well as regularizing towards majority vote labels, indicating that we can substitute one for the other depending on the classification task. Additionally, we notice that as we increase the slack penalty $C$, the performance of our method gets relatively worse. This trend suggests that allowing the constraints to be violated with a small penalty leads to better generalization.

**Constraints**    Running DCWS without constraints, i.e, $\min_\theta \ \left\| f_\theta(X) - \hat{Y}_m \right\|_2^2$ makes use of only one type of data consistency and the majority vote regularization. We see from the results of SST-2 that removing the constraints can significantly reduce the performance of DCWS because the some important information from the weak supervision is not considered.

**Data consistency**    We disabled data consistency by directly solving for the labels of the training data rather than using features of the data to optimize a parametric model. The resulting equation is

$$\min_{\tilde{Y}} \quad \left\| \tilde{Y} - \hat{Y}_m \right\|_2^2 + C \sum_{i=1}^{m} \xi_i$$
$$\text{s.t.} \ \ \boldsymbol{A}\tilde{Y} \leq \boldsymbol{b} + \boldsymbol{\xi} \ \ \text{and} \ \ \boldsymbol{\xi} \geq 0,$$

where $\tilde{Y}$ is the label vector we directly solve for. From Table 5, we see that doing this results in highest performance drop on both datasets. SST-2 achieves a performance that is slightly better than random, while YELP-2 dataset has a 16.8 percentage points drop compared to DCWS. The results emphasizes the need for data consistency in developing label aggregation algorithms. Note that this variation is similar to CLL (Arachie & Huang, 2020).

**Representation**    Using cluster labels as features is another form of data consistent training for label aggregation. We used the same clustering method as described in the synthetic experiments and run DCWS with different numbers of clusters. The results show a significant drop in label accuracies on both datasets. This is in contrast with the superior performance obtained on the synthetic data using cluster labels. This behavior is likely because real data may not be separable or may require a different clustering algorithm to obtain meaningful clusters. Moreover, selecting the appropriate number of clusters adds an additional hyperparameter to the algorithm. We suggest using cluster labels as a representation for DCWS when the data distribution is known and appropriate assumptions can be made for the clustering choice.

**Model**    For all our experiments, DCWS was trained with the same neural network model, however a user can choose a different model architecture depending on their classification task. For example on vision tasks, a user could use a deeper neural network model as $f_\theta(X)$ to get better performance gains. Selecting the best model for each dataset is beyond the scope of our paper so we do no report results of running DCWS on different model architectures. We varied our model slightly by training without dropout and we achieve a drop in performance on both datasets.

**Limitations of DCWS**    Our experiments have shown that DCWS incorporates information from the weak supervision and features of the data to produce quality training labels that are consistent with the data. While our experiments have demonstrated good performance on the datasets we tested, we note some limitations in using our method. DCWS requires a good feature representation of the data to learn meaningful

| Ablation tests | Datasets | |
|---|---|---|
| | SST-2 | YELP-2 |
| DCWS | $0.741\pm 0.002$ | $0.841\pm 0.004$ |
| Without slack | $0.724\pm 0.003$ | $0.829\pm 0.002$ |
| Uniform regularization | $0.740\pm 0.004$ | $0.832\pm 0.002$ |
| Without regularization | $0.739\pm 0.001$ | $0.823\pm 0.004$ |
| Without constraints | $0.671\pm 0.001$ | $0.822\pm 0.001$ |
| Without data consistency | $0.504\pm 0.026$ | $0.667\pm 0.023$ |
| Without dropout | $0.728\pm 0.002$ | $0.828\pm 0.003$ |
| With slack ($C = 0.1$) | $\mathbf{0.750}\pm 0.002$ | $\mathbf{0.844}\pm 0.002$ |
| With slack ($C = 1$) | $0.748\pm 0.003$ | $0.829\pm 0.002$ |
| With slack ($C = 100$) | $0.726\pm 0.003$ | $0.828\pm 0.002$ |
| | | |
| DCWS (10 cluster labels) | $0.560\pm 0.003$ | $0.633\pm 0.000$ |
| DCWS (100 cluster labels) | $0.619\pm 0.000$ | $0.709\pm 0.000$ |
| DCWS (200 cluster labels) | $0.628\pm 0.003$ | $0.705\pm 0.001$ |

Table 5: Results of the ablation study on SST-2 and YELP-2 datasets.

relationships from the data. Using poor features for the data could hurt the performance, because in such settings, consistency with the data is actually bad. DCWS can only work if there is relevant information in the features to relate to the estimated labels. Another limitation is the scalability of our current optimization scheme. We train using gradient descent on the full dataset to accommodate the fact that the constraint is global. This can be computationally intensive for large datasets with very high dimensional features.

Research on aggregating labels for training machine learning systems dates back to early crowdsourcing literature. The first significant work in this research comes from Dawid & Skene (1979). Various improvements and modifications (Welinder et al., 2010; Carpenter, 2008; Gao et al., 2011; Karger et al., 2011; Khetan et al., 2017; Liu et al., 2012; Platanios et al., 2020; Zhou et al., 2015; Zhou & He, 2016) have been made to this original approach, with perhaps the most significant being the algorithm proposed by Zhou et al. (2015). The algorithm has established itself in crowdsourcing literature and is known to be a competitive baseline. While crowdsourcing focuses on generating ground-truth labels from independent human annotators, our work makes no assumption about the error of the individual weak labelers. The weak signals our model takes in can be independent, dependent, or make correlated errors.

Weakly supervised learning algorithms provide another avenue for aggregating labels of training data. These algorithms have gained recent success, partially in part because they allow deep learning models to be trained using only user defined weak signals. A prominent weakly supervised approach is data programming (Ratner et al., 2016), which allows users to combine weak signals via a generative model that estimates the accuracies and dependencies of the weak signals in order to produce probabilistic labels that label the data. Data programming has been implemented as the core of the popular *Snorkel* package for weakly supervised learning. Enhancements have been proposed to the original data programming algorithm (Bach et al., 2019; Ratner et al., 2019; Fu et al., 2020; Chen et al., 2020; 2021), with each method proposing a different learning approach. Our work is related to Snorkel methods in that we combine different weak supervision signals to produce probabilistic labels for the training data. However, unlike Snorkel's methods, we do not make probabilistic modelling assumptions on the joint distribution of the true labels and weak signals. Also, while Snorkel methods are data free and use only the weak signals to estimate the labels of the data, our method is data dependent and use features of the data to make the generated labels consistent with the data. Concurrent to our work, a new weak supervision benchmark has been developed Zhang et al. (2021).

Our work is closely related to constraint-based weak supervision methods Arachie & Huang (2019; 2021; 2020); Mazzetto et al. (2021a;b). These algorithms constrain the possible label space using the weak signals and the error rates of the weak signals and then solve an optimization problem to estimate the labels of the training data. A major advantage of these approach is that they do not make assumptions about the joint distribution of the labeling functions and weak signals. Assumptions about the weak supervision can be hard to obtain in practice and could cause a method to fail on a task. Similar to these methods, we can accept error bound of the weak signals as input when available, and we do not make assumptions on the

joint distribution of the true labels and the weak signals. Unlike these methods, we add slack variables to our algorithm to adaptively loosen our bounds when the estimates of the bounds are incorrect or are fixed as in our setup. Most importantly, the key difference in our approach is that we use input features or data representations to estimate labels that have consistency with the data.

Other algorithms that combine the data with the weak supervision to estimate labels for unlabeled data include (Karamanolakis et al., 2021; Awasthi et al., 2020). These methods leverage labeled data for their label estimation. Unlike these methods, our work is completely weakly supervised and we do not need labeled data to train our model. Our method is also related to other approaches for learning from noisy labels Tanaka et al. (2018); Zheng et al. (2020). These methods focus on correcting for noise in labeled data while our work learns the labels from noisy weak supervision sources.

Classical work on combining weak learners involved using ensemble methods such as boosting (Schapire et al., 2002) to aggregate the learners and create a classifier that outperforms the individual weak learners. The weak learners are trained in a fully supervised learning setting. They differ from our weakly supervised learning approach where we do not have access to the true labels of the data. A more related approach to our setting is a semi-supervised method of Balsubramani & Freund (2015b) that uses unlabeled data to improve accuracy of ensemble binary classifiers. Other notable applications of weak supervision can be found in (Chen et al., 2014; Xu et al., 2014; Bunescu & Mooney, 2007; Hoffmann et al., 2011; Mintz et al., 2009; Riedel et al., 2010; Yao et al., 2010; Halpern et al., 2016; Fries et al., 2019; Saab et al., 2020).

A predominant theme in label aggregation algorithms is estimating accuracy of weak signals without true labels. Methods for estimating accuracy of classifiers without labeled data have been studied in (Collins & Singer, 1999; Jaffe et al., 2016; Madani et al., 2005; Platanios et al., 2014; 2016; Steinhardt & Liang, 2016). These methods assume some knowledge about the true label distribution and then explore statistical relationships such as the agreement rates of the classifiers to estimate their accuracies. Our work is related to these methods in that we use error bounds that provide prior information. Unlike these methods, we do not try to estimate the error bounds by making statistical assumptions on the weak signals.

## 3 Conclusion

We introduced data consistent weak supervision (DCWS), an approach for weakly supervised learning that combines features of the data together with weak supervision signals generate quality labels for the training data. DCWS uses a parametric model and learns parameters that predict the labels of the data. We showed three data representation approaches that can be used in our framework: (i) training with the data itself on the synthetic experiment, (ii) training with embedding of the data, and (iii) training with cluster labels of the training data. Our experiments showed that our data consistency approach significantly outperforms other methods for label aggregation. We also highlight in our experiments that our approach performs well even when the weak signals are very noisy and have low or no coverage. Lastly, we showed in our ablation tests the importance of the different components of our proposed approach. We find that, while the different components of our framework each contribute improvements, data consistency contributes the most to the performance gains achieved by our method.

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

## A  Appendix

## B  Additional Experiments

Similar to the synthetic experiments in the main paper, we randomly define 20 weak signals whose error rates are between $[0.35, 0.45]$. The weak signals have full coverage on the training data. We train data consistent weak supervision (DCWS) with the data itself.

Tables 6a and 6b show the label and test accuracies of the different methods on the synthetic data. Data consistent weak supervision significantly outperforms all competing methods obtaining the highest label accuracy on the data. Surprisingly, DCWS outperforms ALL (another data-dependent method) even though ALL is trained with the true bounds of the weak signals. The inferior performance of ALL can be attributed to the adversary being too powerful and hence choosing worst case labelings that are not data consistent. The performance of the data-free methods are comparable to that of majority voting and only offer slight advantage compared to majority voting. Lastly, we tried another data consistency approach by training DCWS using the cluster labels of the data. We used mini-batch k-means clustering (Sculley, 2010) to obtain the cluster labels. We set the number of clusters to 10 then use a one-hot representation of the cluster labels to train the model. Comparing the two data consistency approaches, running DCWS with cluster labels of the data achieves a higher accuracy score (0.998) than using the data itself. We surmise that the cluster labels in the synthetic setting provide richer information to our algorithm because the data is naturally separable into clusters. This result will not always be the case on real datasets since the data may not be separable and the quality of the cluster labels will depend on the clustering algorithm used. For this reason, our experiments on the real datasets use embedding representations of the data.

## C  Reproducibility

We describe here algorithm and experiment details to help readers reproduce our experiments.

| Method | Label Accuracy |
|---|---|
| DCWS | **0.941** $\pm$ 0.002 |
| FLYINGSQUID | 0.836$\pm$ 0.001 |
| MeTaL | 0.753$\pm$ 0.001 |
| MMCE | 0.752$\pm$ 0.001 |
| CLL | 0.734 $\pm$ 0.001 |
| ALL | 0.497$\pm$ 0.001 |
| Majority Vote | 0.739 $\pm$ 0.001 |

(a) Label Accuracies

| Method | Label Accuracy |
|---|---|
| DCWS | **0.967** $\pm$ 0.002 |
| FLYINGSQUID | 0.887$\pm$ 0.001 |
| MeTaL | 0.834$\pm$ 0.001 |
| MMCE | 0.814$\pm$ 0.001 |
| CLL | 0.823 $\pm$ 0.001 |
| ALL | 0.504$\pm$ 0.001 |
| Majority Vote | 0.787 $\pm$ 0.001 |

(b) Test Accuracies

Table 6: Classification accuracies of the different methods on synthetic data. We report the mean and standard deviation over three trials.

### C.1  Datasets

In this section, we describe the datasets we use in the experiments and the weak supervision we provide to the learning algorithms. Table 7 summarizes the key statistics about these datasets.

| Dataset | No. classes | No. weak signals | Train Size | Test Size | Coverage | Redundancy | Conflict |
|---|---|---|---|---|---|---|---|
| IMDB | 2 | 10 | 29,182 | 20,392 | 0.174 | 1.737 | 0.281 |
| SST-2 | 2 | 14 | 3,998 | 1,821 | 0.103 | 1.445 | 0.202 |
| Yelp | 2 | 14 | 45,370 | 10,000 | 0.177 | 2.475 | 0.358 |
| Fashion-MNIST | 2 | 4 | 1000 | 500 | 1.0 | 4.0 | 0.58 |

Table 7: Summary of datasets and weak signals statistics in our first set of experiments. Coverage is the average number of examples labeled by all the weak signals. Redundancy is the average number of weak signals that label an example in training data. Conflict denotes the fraction of examples covered by conflicting rules in the training data.

We regularize towards majority predictions on the text datasets and uniform distribution on the image dataset.

### C.2  Weak Signals

We provide below the keywords and heuristics we used to generate the weak signals in our experiments. For some signals, we used multiple words since individual word have little coverage in the data. Multiple words signals are represented as nested lists while single words signals are shown as single lists.

**IMDB**    We used 5 positive keywords representing 5 positive signals and 5 negative keywords as 5 negative signals. The positive signals are [good, wonderful, amazing, excellent, great] while the negative signals are [bad, horrible, sucks, awful, terrible].

**SST-2**    Similar to IMDB, however we use 7 positive signals and 7 negative signals that contain nested lists of keywords. The positive signals are

- good, great, nice, delight, wonderful

- love, best, genuine, well, thriller

- clever, enjoy, fine, deliver, fascinating

- super, excellent, charming, pleasure, strong

- fresh, comedy, interesting, fun, entertain, charm, clever

- amazing, romantic, intelligent, classic, stunning

- rich, compelling, delicious, intriguing, smart

while the negative signals are

- bad, better, leave, never, disaster

- nothing, action, fail, suck, difficult

- mess, dull, dumb, bland, outrageous

- slow, terrible, boring, insult, weird, damn

- drag, no, not, awful, waste, flat

- horrible, ridiculous, stupid, annoying, painful

- poor, pathetic, pointless, offensive, silly.

**YELP**  We used the similar weak signals as in SST-2, however we changed the 5th negative weak signals to

- drag, awful, waste, flat, worse.

### C.3 Additional Information on Model Training Experiments

The dataset and weak signals we used for comparing to model training weak supervision methods can be found at the the GitHub page maintained by Awasthi et al. (2020).[2] Detailed descriptions about each dataset task, data size and the weak signals are in Section 3 of the paper Awasthi et al. (2020). Each individual baseline is also explained in Section 3.1 in the paper.

For our experiments, we combine their labeled set with the unlabeled data and use that to train DCWS. The results reported in Table 4 are evaluated on the test set. We included our code to the supplementary material.

---

[2]https://github.com/awasthiabhijeet/Learning-From-Rules

