# OpenReview forum: "Data Consistency for Weakly Supervised Learning"
_TMLR — Rejected by TMLR_

### Review · Reviewer_LiX3 · 2022-08-03

**Summary Of Contributions:**

This paper proposed a method for training a binary classification model based on multiple sources of (soft or hard) noisy labels. The proposed method was evaluated only empirically.

**Broader Impact Concerns:**

The author did not discuss the broader impact.

**Requested Changes:**

The author should greatly improve the writing to highlight the challenge of this problem, the motivation why we should use this approach, the limitations of existing methods, the previous studies that the proposed method is based on, and the contributions of this work.


**Strengths And Weaknesses:**

I did not fully understand this paper (confidence 1), so it is hard for me to assess the value of the proposed method.

The following aspects require attention from the author:

1. The writing and organization should be improved. It is my first time seeing a paper with only three sections: introduction, experiments, and conclusion. The introduction section included background, motivation, problem formulation, related work, approach, and the proposed method. The author just said things like "we did this and we did that" but did not explain why. Which parts are original, and which parts are from previous work? What are the most related studies, and what are their limitations? It's clear that methods for this problem do exist, so why do we need yet another method? The proposed method is not well contextualized, and the motivation is not convincing.
2. The meaning of "data consistency" is unclear. Consistency has a specific and well-accepted definition in statistical learning, but it seems the meaning of "data consistency" has nothing to do with it. It is okay (but risky) if the author wants to coin a new term, but the author should explain its meaning better. It seems to me that it just means that *the label depends on the input* (?). It sounds weird because it is the case for all supervised learning models.
3. The author said "noisy labels, i.e., weak signals" multiple times and implied that they have the same meaning, but it is not true. Weak supervision refers to a much larger range of supervision, including noisy label, partial label, similarity, comparison, summary statistics, multi-instance, etc. Thus, I think "weakly supervised learning" in the title is imprecise and overclaiming. A more detailed title can make this paper reach its target audience better.
4. "No assumptions" is not always a good thing because it usually means "no theory", just as in this paper.

---

### Review · Reviewer_DCvU · 2022-08-04

**Summary Of Contributions:**

The paper presents a new method for weakly supervised learning. This work is a direct extension of [1]. The key novelty of the approach compared to [1] work lies in the idea to use features of the unlabeled data to generate labels for training, so called "data consistency".  The authors achieve this by defining a constrained objective function which uses input features of the unlabeled data and weak supervision signals thereby assuming that the labels are a function of the input features. Additionally, authors introduce slack variables to adaptively loosen error bounds.

[1] Arachie & Huang. "Constrained Labeling for Weakly Supervised Learning". UAI '21

**Broader Impact Concerns:**

There are no ethical concerns.

**Requested Changes:**

- The authors claim that the slack penalty helps when error bounds cannot be accurately estimated or when they are too loose/too tight but there should be an analysis and experiments that clearly demonstrate this.
-  There should be a related work section. Currently, related work is mentioned in the ablation study.
- Given that the gradient descent is applied to the full dataset, how does the method scale to large datasets? Can you compare training time between the methods as well as the dependence of the training time on the dataset size?
- How does the method compare to Mazzetto et al ICML 2021 and Arachie & Huang UAI '21?
- Can you provide consistent comparison of baselines, i.e., all baselines on all datasets?
- Why are results for ALL missing in Tables 2 and 3?
- Why is slack penalty set to 10 and how is it determined?
- How are parameters for baseline methods determined? Do you use same backbone networks in the experiments? Is the network architecture of the label model same as the model used to test the generalizablity of the algorithm? All of that should be clear.
- The authors claim to outperform semi-supervised learning methods but there are no SOA SSL methods included as the baselines. This needs to be changed.

Minor:
- References are messed up. Most of the arXIv references are already published.
- Additional details of the baselines are said to be included in the appendix but are not there.

**Strengths And Weaknesses:**

Strengths:
- the proposed method allows that the weak supervision signals abstains on data subsets
- introducing slack penalty helps in cases when the error bounds cannot be accurately estimated which is often the case in practice
- slack penalty and data consistency introduced in this work lead to performance gains

Weaknesses:
- the paper is not well written
- the clear motivation behind the work is missing. The main novelty seems to be in using input features jointly with weak supervision signal, but it is not explained why is it important to use input features and how does this approach makes substantial contribution compared to existing works
- compared to previous works, there is no theoretical analysis on the error rate
- experimental setup is unclear and the results are not convicing. The authors compare to one set of baselines on one set of datasets, and to another set of baselines on the other set of datasets. In the first set are methods that do not consider the data for label aggregation, but ALL is included even though it is not in that group. The work is related to Arachie & Huang, UAI '21 and Mazzetto et al ICML '21, but authors do not compare to these methods. Furthermore, authors claim to outperform semi-supervised methods, but SOA SSL methods are not included in the experiments.

---

### Review · Reviewer_UoJW · 2022-08-06

**Summary Of Contributions:**

The paper proposes an optimization formulation for the problem of trianing under weakly-supervised signals. Particularly, the formulation aims to find a label model whose predictions can match the expected error rates of the weakly-supervised signals. To avoid estimating the expected error rates of weakly supervised signals, the paper uses slack variables and proposes to solve the optimization using the lagrangian multiplier. The paper empirically demonstrates that the proposed method can achieve better labeling accuracy on the training dataset and prediction accuracy on the unseen testing dataset.

**Broader Impact Concerns:**

N/A.

**Requested Changes:**

Please address the issues/questions in the weakness part mentioned above.

**Strengths And Weaknesses:**

Strengths:
1. The paper proposes an optimization formulation for the training problem under weakly-supervised signals. The formulation uses slack variables to avoid estimating the weak signals' expected error rates.

2. The experiments show that the proposed method can give superior results for the given settings.

Weaknesses:
1. The paper's contribution is limited as it heavily depends on the previous work of [Arachie & Huang (2020)]. The major change in the paper compared to the previous work is using the slack variable to avoid estimating the weak signals' expected error rates.

2. As mentioned in the paper, the proposed optimization method does not scale well as it requires calculating the full gradients over all data samples to ensure that the constraints are always satisfied.

3. I think it requires some more effort in the empirical evaluation part to further demonstrate the proposed methods' effectiveness. Firstly, the testing datasets are relatively small and simple. I suggest showing results on more complicated datasets. Secondly, the setup mostly has only two classes. The paper argues that the proposed method can work for multi-class cases, but the empirical evidence is limited. Thirdly, the weakly-supervised signals are all synthesized. I suggest trying realistic datasets with real-world weak supervision (e.g. the webvision dataset).

4. I think the weakly-supervised signal learning problem is closely related to noisy-label learning. There is some discussion about noisy label learning, but I think more is required to elaborate on the connections/differences between the two settings. A possible approach that utilizes noisy label learning methods to solve the weakly-supervised signal learning problem is to first generate some labels in consensus (e.g., through majority voting), and then apply noisy label learning algorithms to the generated labels. Therefore, it is also valuable to empirically test against the state-of-the-art noisy label learning methods (few noisy-label learning methods are tested in the paper, but they are relatively out-dated).

---

### Decision · Action_Editors · 2022-09-09

**Recommendation:** Reject

**Comment:**

First, I would like to thank the authors for their submission. In short, the authors proposed a weak supervision technique that relies on solving a particular constrained optimization problem to enable adaptively loosening a set of error bounds on the weak supervision signals. This is in contrast to a number of existing weak supervision techniques. This approach shows good performance in a number of experimental settings.

While the paper in its current state already offers a number of interesting nuggets, the consensus of the reviewers is that it faces several issues, and I agree, leading me to vote for rejection. The common threads identified by the reviewers fall into two categories. First, there is insufficient evidence for some of the claims made, in particular the idea that the technique is state-of-the-art in weak supervision. Second, writing clarity did not meet the bar. However, the reviewers raised a number of points and suggestions that would significantly improve a future version of the paper and mitigate these concerns.

Two of the reviewers struggled to find sufficient evidence of the claims from both an empirical and analytical points of view. I tend to agree with this concern. It could potentially be mitigated with more extensive evaluations or further analysis, qualitative or quantitative. The strength of the paper rests on the empirical evaluations, and here there are some solid results. The authors produced a synthetic data case that matched expectations and produced solid improvements. The reviewers believed that more extensive experimental evaluation is necessary. One way to do this is to use the WRENCH benchmark, which allows for much broader comparison in terms of datasets, modalities, and baselines. Indeed this would enable evaluating the state-of-the-art claim in a reasonable way.

Another common concern was the writing of the paper. While in some places the writing was crisp, as a whole the organization lacked sufficient clarity. A more extensive background that situates this work and provides further detail into the burgeoning weak supervision literature would be appreciated. There are at least a few additional existing works on weak supervision that are also data-dependent, and a discussion and differentiation from these works would serve the current paper well.

I am optimistic about the overall direction and hope the authors can use the feedback from the reviewers for an improved future work.